# Morphological and Physiological Mechanisms of Melatonin on Delaying Drought-Induced Leaf Senescence in Cotton

**DOI:** 10.3390/ijms24087269

**Published:** 2023-04-14

**Authors:** Kai Yang, Hongchun Sun, Mengxing Liu, Lingxiao Zhu, Ke Zhang, Yongjiang Zhang, Anchang Li, Haina Zhang, Jijie Zhu, Xiaoqing Liu, Zhiying Bai, Liantao Liu, Cundong Li

**Affiliations:** 1State Key Laboratory of North China Crop Improvement and Regulation, Key Laboratory of Crop Growth Regulation of Hebei Province, College of Agronomy, Hebei Agricultural University, Baoding 071001, China; auhykday@163.com (K.Y.);; 2Cotton Research Institute, Hebei Academy of Agriculture and Forestry Sciences, Shijiazhuang 050051, China; 3Institute of Cereal and Oil Crops, Hebei Academy of Agriculture and Forestry Sciences, Shijiazhuang 050051, China

**Keywords:** melatonin, leaf senescence, cotton, drought, photosynthesis, antioxidant, gene expression

## Abstract

Leaf senescence reduces the photosynthetic capacity of leaves, thus significantly affecting the growth, development, and yield formation of cotton. Melatonin (MT) is a multipotent substance proven to delay leaf senescence. However, its potential mechanism in delaying leaf senescence induced by abiotic stress remains unclear. This study aimed to explore the effect of MT on delaying drought-induced leaf senescence in cotton seedlings and to clarify its morphological and physiological mechanisms. Drought stress upregulated the leaf senescence marker genes, destroyed the photosystem, and led to excessive accumulation of reactive oxygen species (ROS, e.g., H_2_O_2_ and O_2_^−^), thus accelerating leaf senescence. However, leaf senescence was significantly delayed when 100 μM MT was sprayed on the leaves of the cotton seedlings. The delay was embodied by the increased chlorophyll content, photosynthetic capacity, and antioxidant enzyme activities, as well as decreased H_2_O_2_, O_2_^−^, and abscisic acid (ABA) contents by 34.44%, 37.68%, and 29.32%, respectively. MT significantly down-regulated chlorophyll degradation-related genes and senescence marker genes (*GhNAC12* and *GhWRKY27/71*). In addition, MT reduced the chloroplast damage caused by drought-induced leaf senescence and maintained the integrity of the chloroplast lamellae structure under drought stress. The findings of this study collectively suggest that MT can effectively enhance the antioxidant enzyme system, improve photosynthetic efficiency, reduce chlorophyll degradation and ROS accumulation, and inhibit ABA synthesis, thereby delaying drought-induced leaf senescence in cotton.

## 1. Introduction

Leaf senescence is the final process of leaf development and leads to the loss of crop organ function [1]. Senescence plays an essential role in plant development through the recycling of many important nutrients. It is easily induced by various stress factors, such as drought, salinity, high or low temperature, darkness, nutrient deficiency, and pathogen infection [2]. Premature leaf senescence is a degenerative physiological process that affects crop yield because it reduces photosynthetic efficiency and nutrient accumulation [3,4]. Cotton (*Gossypium hirsutum* L.) is the most essential fiber crop and is a valuable oil crop worldwide [5]. Premature leaf senescence significantly restricts cotton yield and quality [6]. In China, the yield reduction can reach 225–300 kg/hm^2^ [7]. Premature leaf senescence is closely associated with drought stress (DS) during the early growth of cotton [8]. It is attributed to cotton’s extreme sensitivity to water shortage because of its origin in tropical regions with abundant water [9].

Leaf senescence is correlated with various biochemical changes [10]. Chlorophyll degradation is the most significant characteristic of natural leaf senescence caused by stress [11]. The senescence process is closely associated with the destruction of the chloroplast structure, an increase of membrane lipid peroxide content, and a significant decrease in the net photosynthetic rate (Pn) and photosystem II (PSII) photochemical efficiency [12,13]. Excessive accumulation of reactive oxygen species (ROS) is also an essential feature of leaf senescence [14]. For example, there is excessive accumulation of superoxide anions (O_2_^−^) and hydrogen peroxide (H_2_O_2_) in the dark-induced senescence of grape (*Vitis vinifera* L.) leaves [15], heat-induced senescence of tomato (*Solanum lycopersicum* L.) leaves [16], and postharvest storage of cabbage (*Brassica rapa* ssp. *parachinensis*) leaves [17]. In addition, various plant hormones, such as abscisic acid (ABA), auxin (IAA), cytokinin (CTK), and gibberellin (GA), participate in regulating leaf senescence [18]. Notably, the content of endogenous ABA is higher in senescent leaves, while IAA, CTK, and GA delay leaf senescence [19,20,21,22]. Leaf senescence is also reflected by the differential expression of senescence marker genes and NAC and WRKY transcription factors [23,24].

Leaf senescence and accelerated shedding are often manifestations of plant drought tolerance in the wild because they help plants to maintain water balance [25,26], thereby increasing their survival chances under DS [27]. However, leaf senescence in field crops is closely associated with yield reduction [28]. Abiotic stress-induced leaf senescence significantly affects crop yield and quality [29]. Notably, the area and degree of arid and semi-arid regions are increasing because of global warming, insufficient rainfall, and uneven rainfall distribution [30]. The loss of crop yield caused by drought-induced leaf senescence thus exceeds losses attributed to other abiotic stress factors [31]. Delaying leaf senescence can enhance drought resistance and improve photosynthesis, which is the basis of high yields [32]. Exploring methods to delay leaf senescence are thus prerequisites for maintaining and enhancing agricultural production under abiotic stress conditions.

Melatonin (N-acetyl-5-methoxytryptamine, MT) is a multi-effect small molecule indole widely found in animals, prokaryotes, and plants [33]. MT can regulate plant growth and development and is a powerful antioxidant and ROS scavenger that activates seed germination [34], regulates root development [35], promotes plant growth [36], and delays leaf senescence [37]. MT removes excess ROS and prolongs the function of leaves during leaf senescence [38]. The root application of MT delays drought-induced senescence of apple (*Malus domestica* Borkh.) leaves by scavenging ROS and reducing chlorophyll degradation [39]. MT delays dark-induced ryegrass (*Lolium perenne* L.) senescence by activating the ROS scavenging pathway and down-regulating senescence marker genes [40]. Moreover, MT delays dark-induced cucumber (*Cucumis sativus* L.) leaf senescence by improving photosynthetic efficiency, protecting PSII activity, and maintaining chloroplast ultrastructure [41]. The level of MT in a plant is correlated with the content of hormones regulating leaf senescence [42]. Pretreatment of tomatoes with MT can delay high temperature-induced leaf senescence by inhibiting ABA formation and increasing the GA content [16]. Similarly, foliar spraying of tobacco (*Nicotiana tabacum* L.) with MT enhances its resistance to dehydration-induced leaf senescence by regulating the phytohormone-gene regulatory networks, mainly up-regulation of the IAA content and down-regulation of the ABA content [43]. MT also alleviates heat-induced leaf senescence by increasing the CTK content (t-ZR and iPA) and endogenous MT in perennial ryegrass [44].

In conclusion, MT has a huge potential to delay leaf senescence [45]. However, it remains unclear how it regulates the morphology, physiology, biochemical, and the ultra-microstructure of cotton leaves to alleviate drought-induced leaf senescence. This study mainly focused on the mechanisms of MT in regulating photosynthesis, membrane damage, root morphology, chloroplast structure, and ROS dynamic balance of cotton seedlings during drought-induced leaf senescence. The study also probed the MT mechanism in regulating the antioxidant oxidase defense system and endogenous hormone levels. The findings of this study provide a theoretical basis for applying MT to alleviate drought-induced premature leaf senescence in cotton.

## 2. Results

### 2.1. Effects of MT on the Growth and Development of Cotton Seedlings under DS

Cotton seedlings were irrigated with PEG-6000 for 14 days to simulate DS and subsequent leaf senescence. DS inhibited the growth of cotton seedlings. However, this phenomenon was alleviated by spraying exogenous MT (Figure 1A). Plant height, stem diameter, and leaf area were decreased by 19.69%, 16.36%, and 24.49% at 14 days after treatment (DAT), respectively, under DS compared to the corresponding parameters of CK seedlings. Notably, spraying MT increased plant height, stem diameter, and leaf area by 16.71%, 3.70%, and 20.82% compared to DS (*p* < 0.05) (Figure 1B–D). The SPAD value of leaves increased by 5.45% after spraying MT compared to DS (*p* < 0.05) at 14 DAT (Figure 1E), indicating that MT reduced and prevented chlorophyll degradation.

Compared to CK, DS treatment significantly reduced the fresh shoot, dry shoot, fresh root, and dry root weights of cotton seedlings (Figure 2A–D). Compared to DS, MT + DS treatment increased the fresh shoot and root weights by 24.29% and 50.14% and the dry shoot and root weights by 9.64% and 12.27%, respectively (*p* < 0.05). DS inhibited the root growth of cotton seedlings (Figure 2E–H). Compared to CK, DS reduced the root length, root surface area, and root volume by 57.18%, 58.59%, and 58.75% (*p* < 0.05), respectively. However, MT + DS treatment increased these indicators by 17.57%, 31.73%, and 41.30% (*p* < 0.05), respectively, compared to the corresponding indicators under DS treatment, indicating that MT can alleviate the growth inhibition of both the cotton shoots and the roots during drought-induced senescence.

### 2.2. Effects of MT on the Photosynthetic Characteristics

Compared to CK, the Pn under DS treatment decreased by 39.92% and 61.71% (*p* < 0.05), while the Pn under MT + DS treatment decreased by 25.68% and 43.19% (*p* < 0.05) at 7 DAT and 14 DAT, respectively (Figure 3A). There was a significant decrease in transpiration rate (Tr), intercellular CO_2_ concentration (Ci), and stomatal conductance (Gs) under DS treatment compared to CK (Figure 3B–D). Compared to DS, Tr, Ci, and Gs under MT + DS treatment increased by 63.68%, 17.80%, and 62.23%at 14 DAT (*p* < 0.05), respectively. In addition, DS significantly decreased the CO_2_ assimilation rate after fitting the parameters of the light response curve. Compared to DS, MT + DS treatment significantly improved the photosynthetic capacity of cotton, especially under high light levels (Appendix A). Compared to DS, the maximum net photosynthetic rate (P_max_), light compensation point (LCP), and respiratory rate (R_d_) of MT increased by 35.74%, 99.67%, and 154.13% (*p* < 0.05), respectively (Table 1). In summary, MT alleviated the decrease of photosynthetic capacity during drought-induced leaf senescence.

DS treatment significantly decreased the maximum photochemical efficiency (Fv/Fm). However, Fv/Fm reduction under MT + DS treatment was significantly lower than under DS treatment but increased by 1.82% compared to that of DS at 14 DAT (*p* < 0.05) (Figure 4A). DS treatment also significantly decreased the actual photochemical quantum yield of PSII (ΦPSII) and apparent electron transfer efficiency (ETR) compared to CK (Figure 4B,C). Non-photochemical quenching (NPQ) showed an increasing trend during the process of drought-induced senescence. Of note, NPQ under DS treatment was significantly higher than that under CK and MT + DS treatments at 14 DAT (Figure 4D). These data show that MT can effectively increase the Fv/Fm, ΦPSII, and ETR of cotton seedlings to alleviate the photosynthetic inhibition induced by DS.

### 2.3. Effects of MT on Relative Water Content (RWC) and Malondialdehyde (MDA) Content

The leaves of cotton seedlings subjected to DS using PEG-6000 were gradually dehydrated. Notably, the RWC of the leaves decreased from 90.51% at 0 DAT to 75.67% at 14 DAT (Figure 5A). In contrast, the content of MDA in the leaves increased rapidly, with the MDA content at 14 DAT being 3.57 times higher than that at 0 DAT (Figure 5B). Compared to DS, MT + DS treatment significantly increased the RWC of leaves and maintained it at 84.93% at 14 DAT. However, MT + DS treatment significantly reduced the MDA content by 86.42% (*p* < 0.05). This phenomenon shows that MT can delay leaf senescence by maintaining a high RWC of leaves and reducing the MDA content in cotton seedlings.

### 2.4. Effects of MT on the Antioxidant Enzyme Activity

The activities of superoxide dismutase (SOD), peroxidase (POD), catalase (CAT), and ascorbate peroxidase (APX) under DS and MT + DS treatments showed a similar change pattern; their activities first increased and then decreased. However, there were no apparent changes in enzyme activity in cotton seedlings under CK treatment (Figure 6A–D). Compared to CK, the activities of SOD, POD, and APX under DS treatment were significantly increased, while the activities of several enzymes tended to decrease with the aggravation of drought-induced leaf senescence. Compared to DS treatment, MT + DS treatment increased the activities of SOD, POD, and APX by 6.72%, 5.61%, and 50% at 14 DAT (*p* < 0.05), respectively (Figure 6A,B,D). The activity of CAT under DS and MT + DS treatments was significantly lower than that of CK treatment at 14 DAT. However, CAT activity under MT + DS treatment was 22.82% higher than under DS treatment (*p* < 0.05) (Figure 6C). This phenomenon indicated that CAT activity was more sensitive to leaf senescence. However, spraying MT effectively increased the activity of CAT. MT can thus delay drought-induced leaf senescence by increasing antioxidant enzyme activity.

### 2.5. Effect of MT on the ROS Content

Figure 7A,B shows the level of H_2_O_2_ and O_2_^−^ in cotton leaves detected through 3,3-diaminobenzidine (DAB) and nitro-blue tetrazolium (NBT) staining methods. Notably, brown and blue spots were spread over the entire leaf area under DS treatment but only in a small area under MT + DS treatment. Histochemical staining showed that MT significantly decreased the H_2_O_2_ and O_2_^−^ contents during drought-induced senescence. H_2_O_2_ and O_2_^−^ contents in cotton leaves exhibited an increasing trend as the DS continued. The levels of H_2_O_2_ and O_2_^−^ under DS treatment increased by 4.20 times and 1.47 times compared to those of CK-treated seedlings at 14 DAT (*p* < 0.05) (Figure 7C,D). However, the contents of H_2_O_2_ and O_2_^−^ in MT-treated seedlings were reduced by 34.44% and 37.68% (*p* < 0.05), respectively, compared to those under DS treatment (Figure 7C,D). These results indicated that MT could effectively inhibit ROS accumulation in cotton seedlings under DS, thereby delaying leaf senescence.

### 2.6. Effects of MT on the Endogenous Hormone Content

The IAA level of seedlings under CK treatment gradually increased with the growth of seedlings. However, the IAA level of DS-treated seedlings decreased by 61.76% compared to the corresponding IAA level in CK seedlings at 14 DAT. In the same line, the IAA level of MT-treated seedlings increased by 110.32% compared to that of DS-treated seedlings (*p* < 0.05) (Figure 8A). Compared to CK seedlings, the ABA level of DS-treated seedlings increased by 61.69% and 110.19% at 7 DAT and 14 DAT (*p* < 0.05), respectively. Compared to DS-treated seedlings, the ABA level of MT-treated seedlings decreased by 24.57% and 29.32% at 7 DAT and 14 DAT (*p* < 0.05), respectively (Figure 8B). The contents of GA_3_ and iPA gradually decreased with the aggravation of drought-induced senescence. The contents of GA_3_ and iPA in DS treatment reduced by 15.76% and 34.98% at 14 DAT, respectively, compared to the corresponding contents in CK seedlings. In contrast, the contents of GA_3_ and iPA in MT-treated seedlings increased by 26.15% and 87.42% at 14 DAT, respectively, compared to the corresponding contents in DS-treated seedlings (Figure 8C,D). In the same line, the endogenous melatonin content of MT-treated seedlings increased by 33.14% at 14 DAT compared to the corresponding contents in DS-treated seedlings (Figure 8E). In summary, MT delayed drought-induced leaf senescence by increasing the levels of IAA, GA_3_, and iPA and reducing the level of ABA in the leaves.

### 2.7. Effects of MT on the Ultrastructural Alterations in the Chloroplast

In this study we observed the ultrastructure of the mesophyll cell structure during drought-induced leaf senescence to determine the effect of MT on the ultrastructural alterations in the chloroplast (Figure 9A–F). The chloroplasts under CK treatment had a regular oval shape, which was neatly and tightly attached to the cell wall (Figure 9A,D). In contrast, the structure of mesophyll cells in leaves under DS treatment was severely damaged. Of note, plasmolysis occurred despite their morphology not being significantly deformed (Figure 9B). In addition, drought-induced leaf senescence reduced the length, increased the width, and decreased the aspect ratio of the chloroplasts. The number of basal grains in the chloroplasts was significantly reduced, and the lamella structure was disordered, tending to be degraded and broken (Figure 9E). However, the mesophyll cells of the leaves under MT + DS treatment did not have plasmolysis, and the shape of the chloroplasts was relatively regular oval, which was basically similar to the shape of the chloroplasts under CK treatment. The grana lamellar structure was complete, and the arrangement was relatively orderly (Figure 9C,F). These results demonstrated that drought-induced leaf senescence caused severe damage to the cell structure. Of note, MT + DS treatment significantly alleviated the membrane lipid damage caused by leaf senescence and maintained the basic structure of the chloroplasts.

### 2.8. Effects of MT on the Transcript Abundance of Chlorophyll Degradation Genes, Senescence Marker Genes, and Hormone Synthesis Genes during Leaf Senescence

This study determined the mRNA levels of chlorophyll degradation genes, senescence-related genes, and hormone synthesis genes in the cotton leaves at 14 DAT to further verify the role of MT in delaying drought-induced leaf senescence (Figure 10). Notably, MT + DS treatment up-regulated the large subunit of Rubisco, ribulose-1, 5-bisphosphate carboxylase/oxygenase (*GhRBCL*) but down-regulated *non-yellow coloring 1* (*GhNYC1*) (Figure 10A,B). This phenomenon suggests that MT potentially inhibits chlorophyll degradation by increasing the level of *GhRBCL* and decreasing the level of *GhNYC1*. *WRKY27/71* and *NAC12* are positive leaf senescence regulators significantly upregulated during drought-induced senescence. However, their expression was significantly reduced by MT + DS treatment (Figure 10C–E), effectively delaying leaf senescence.

The expression level of hormone synthesis genes was generally consistent with that of endogenous hormone content. The expression levels of *flavin-binding monooxygenase family protein* (*GhYUC5*), *gibberellin 3-oxidase 2* (*GhGA3ox2*), and *tRNAisopentenyltransferase 2* (*GhIPT2*) decreased significantly under DS treatment but increased significantly under MT + DS treatment compared to the corresponding levels under CK treatment. Senescence significantly upregulated the expression of the ABA synthesis gene *nine-cis-epoxycarotenoid dioxygenase 2* (*GhNCED2*). However, MT effectively inhibited the up-regulation of *GhNCED2* (Figure 10F–I) and the down-regulation of *serotonin N-acetyltransferase* (*GhSNAT*) caused by leaf senescence. In addition, a Pearson’s coefficient test on the levels of endogenous IAA, ABA, GA_3_, iPA, and MT and the expression levels of related synthetic genes was performed to understand the relationship between hormone biosynthesis (Appendix A). There was a significant positive correlation between IAA, GA_3_, iPA, MT, *GhYUC5*, *GhGA3ox2*, *GhIPT2*, and *GhSNAT* and between ABA and *GhNCED2*. This phenomenon indicated that the content of endogenous hormones is closely associated with the expression of its related synthetic genes. Moreover, spraying MT can inhibit the synthesis of ABA and play a positive regulatory role in IAA, GA_3_, and iPA.

### 2.9. Correlation and Principal Component Analyses

A Pearson correlation analysis of 21 representative traits was performed to identify their correlation (Appendix A). SPAD, Pn, Fv/Fm, and RWC were negatively correlated with MDA, H_2_O_2_, O_2_^−^, and ABA but positively correlated with CAT, IAA, GA_3_, iPA, MT, and *RBCL*. *NYC1*, *WRKY27*/*71,* and *NAC12* were positively correlated with MDA, H_2_O_2_, O_2_^−^, and ABA but negatively correlated with SPAD, Pn, Fv/Fm, RWC, CAT, and *RBCL*.

The contribution rate of principal component 1 was 74.4%. *NYC1*, *WRKY27*, *WRKY71*, *NAC12*, MDA, H_2_O_2_, O_2_^−^, and ABA were close to the positive direction of the x-axis, while RWC, CAT, SPAD, Pn, Fv/Fm, and *RBCL* are in the positive direction of the x-axis negative direction (Appendix A). The contribution rate of principal component 2 was 18.1% and mainly included APX, POD, and SOD. The total contribution rate of 92.9% explained most of the information and revealed that MT significantly affected the expression of senescence marker genes, ROS, and ABA levels, thus regulating leaf senescence.

## 3. Discussion

Potted vermiculite was used in this study to determine the changes in seedling biomass, photosynthesis, fluorescence parameters, antioxidant enzyme activity, endogenous hormone content, chloroplast ultrastructure, and senescence marker genes expression during drought-induced cotton leaf senescence. This study aimed to test the hypothesis that MT acts morphologically and physiologically to delay cotton leaf senescence.

The phenotypic yellowing and wilting of leaves during leaf senescence are attributed to the degradation of chlorophyll and the increase of membrane lipid peroxidation, thus impeding normal photosynthesis [46]. Pn, MDA, and chlorophyll contents are usually used as effective physiological indicators of leaf senescence [47]. Herein, DS treatment disordered the lamellar structure of the chloroplast, significantly decreased SPAD, Pn, and Fv/Fm, significantly increased MDA, H_2_O_2_, and O_2_^−^ content, and upregulated the expression of senescence marker genes and chlorophyll degradation genes at 14 DAT. These changes showed that 15% PEG-6000 simulated drought, thereby causing leaf senescence at 14 DAT. Notably, the specific symptoms of drought-induced leaf senescence were chlorophyll degradation, chloroplast ultrastructure changes, a sharp decrease in photosynthetic capacity, a significant increase in membrane lipid peroxidation, excessive accumulation of ROS, and differential expression of senescence marker genes [43,48].

### 3.1. MT Delays Leaf Senescence by Regulating Phytohormone Biosynthesis and Expression of Senescence Marker Genes

Plant hormones are essential in regulating leaf senescence [18]. Notably, plant hormone signal transduction is vital in regulating plant response to drought and water shortage [49]. MT interacts with plant hormones and plays a beneficial role in the aging process and stress response [50]. ABA is the promoter of senescence under biotic and abiotic stress [51]. Dong et al. [52] reported that the ABA content increased gradually during cotton leaf senescence. Interestingly, MT significantly inhibits ABA accumulation and expression of ABA biosynthesis genes (*NCED1* and *NCED2*) in plants [16]. In this study, MT alleviated drought-induced leaf senescence by reducing ABA levels (Figure 7B). Subsequent qPCR assays confirmed that MT decreased the expression of the ABA synthesis gene *GhNCED2* (Figure 2G). In contrast, IAA, GA_3_, and CTK play active roles in delaying leaf senescence. MT can alleviate the inhibition of senescence by regulating IAA, GA_3_, and CTK biosynthesis [53]. The findings of this study were consistent with previous studies affirming that MT delays leaf senescence under DS by up-regulating IAA, GA_3_, and iPA biosynthesis-related genes (*GhYUC5*, *GhGA3ox2*, and *GhIPT2*) and increasing the endogenous IAA, GA_3_, and iPA content (Figure 7).

The synergistic regulation of these hormones is closely associated with the increase of endogenous MT levels through the addition of exogenous MT (Figure 8E). The increase in endogenous MT is attributed to the absorption of exogenous MT or the induction of MT biosynthesis genes in cotton leaves under DS. MT pretreatment increases the expression of MT biosynthesis genes, such as *NtSNAT* and *acetylserotonin O-methyltransferase 1* (*NtASMT1*), in drought-induced senescence in tobacco leaves [43]. In this study, the MT biosynthesis gene *GhSNAT* was significantly down-regulated in drought-induced leaf senescence (Figure 10J). This phenomenon demonstrated that exogenous MT can affect the level of endogenous MT, thereby upregulating the levels of IAA, GA_3_, and iPA and inhibiting ABA biosynthesis, thus delaying leaf senescence.

Leaf senescence is regulated by hormones and the expression of senescence marker genes [54]. *GhNAC12* is a candidate gene for the early senescence of upland cotton varieties [55]. Overexpression of *GhWRKY27* can promote leaf senescence by reducing the chlorophyll content. *GhWRKY71* plays a positive role in regulating leaf senescence [56]. Herein, DS treatment significantly upregulated senescence marker genes. However, MT significantly inhibited the expression of *GhWRKY27/71* and *GhNAC12* at 14 DAT (Figure 1C–E), suggesting that MT delays leaf senescence by inhibiting the expression of senescence marker genes, such as *GhWRKY27/71* and *GhNAC12*.

### 3.2. MT Delays Leaf Senescence by Inhibiting Chlorophyll Degradation

Chlorophyll degradation is a sign of the initiation of leaf senescence [57]. Drought-induced leaf senescence accelerates chlorophyll degradation. In this study, MT maintained the SPAD of cotton seedlings at a high level (Figure 1D). A previous study postulated that the exogenous application of MT also alleviates chlorophyll degradation during drought-induced leaf senescence in creeping bentgrass (*Agrostis stolonifera* L.) [58]. Chlorophyll and chlorophyll-degrading enzymes are distributed in different parts of the chloroplast. The structural damage of chloroplasts during senescence leads to their combination, thereby accelerating chlorophyll degradation [11]. In addition, exogenous MT can alleviate chlorophyll degradation by down-regulating the expression of critical genes, such as *non-yellow coloring* (*NYC*) and *pheophorbide a oxygenase* (*PAO*)*,* thereby delaying the leaf senescence of tomatoes [16]. This study confirmed that MT could inhibit the expression of the chlorophyll degradation gene *NYC1* (Figure 10B). MT-mediated regulation of the expression of critical genes in chlorophyll degradation is thus a potential mechanism for delaying leaf senescence during abiotic stress-induced senescence.

Importantly, this study also demonstrated that MT can maintain a relatively intact chloroplast structure during drought-induced leaf senescence (Figure 9). Notably, DS treatment gradually shortened the chloroplast length and ruptured the chloroplast membrane, matrix lamella, grana, and thylakoid membrane [59]. However, there was a significant reduction in chloroplast damage caused by DS under MT + DS treatment.

### 3.3. MT Delays Leaf Senescence by Regulating Redox Homeostasis

ROS generation and scavenging ability in plants tend to be unbalanced during leaf senescence. The biological macromolecules are also damaged, leading to membrane lipid peroxidation, cell senescence, and apoptosis [60]. Spraying MT can significantly reduce ROS accumulation and membrane lipid peroxidation in maize (*Zea mays* L.) [37], soybean (*Glycine max* L.) [61], and cotton [62]. In this study, the levels of MDA, H_2_O_2_, and O_2_^−^ increased significantly with the aggravation of leaf senescence (Figure 5 and Figure 7). Notably, there was a significant increase in MDA, H_2_O_2_, and O_2_^−^ under MT + DS treatment compared to DS treatment, suggesting that MT can act as an antioxidant to alleviate the damage caused by ROS accumulation in cotton leaves, thereby delaying leaf senescence.

Antioxidant enzymes participate in important pathways that aid plants in removing excess ROS. The defense system of antioxidant enzymes in plants is activated during abiotic stress-induced leaf senescence, causing the activities of SOD, POD, CAT, and APX to increase [63]. The exogenous application of MT significantly increases the activities of antioxidant enzymes in marjoram (*Origanum marjorana* L.) [64]. In this study, MT significantly increased the activity of the antioxidant enzymes during senescence (Figure 6). Compared to other antioxidant enzymes, CAT was more sensitive to DS and exogenous melatonin. This result is inconsistent with the inference of Tan et al. [17] that CAT has a delayed response to melatonin, possibly because of organelle specificity or other unknown genetic factors. The increase in the activity of antioxidant enzymes indicated that MT alleviates the oxidative damage of cells by strengthening the defense system of antioxidant enzymes, thus delaying leaf senescence.

### 3.4. MT Delays Leaf Senescence and Promotes Plant Growth by Increasing Photosynthetic Efficiency

A typical feature of leaf senescence is a decrease in photosynthesis caused by changes in the chloroplast structure and function [65,66]. The reduction of photosynthesis induced by premature senescence causes a huge loss in cotton yield and quality in field production [67]. MT can protect the photosynthetic system, thereby improving the Pn, Tr, and Gs during senescence [68]. Herein, the Pn of cotton seedlings was significantly reduced during drought-induced leaf senescence. However, MT-treated seedlings had higher Pn, Gs, Tr, and Ci (Figure 3). MT-mediated protection of the leaf chloroplast structures from oxidative damage is the primary reason for the melatonin-mediated improvement of photosynthesis during drought-induced leaf senescence [69]. Kong et al. [70] reported that the expression level of *RBCL* gradually decreased with the senescence of cotton leaves. Of note, a decrease in the *RBCL* level may lead to the degradation of Rubisco, thus reducing the Pn. RBCL protein levels decrease during darkness-induced cucumber leaf senescence. However, MT + DS treatment has been reported to increase the stability of the RBCL protein to maintain a high Pn in cucumber leaves, thus delaying senescence [41]. Herein, MT effectively inhibited the downregulation of *RBCL* during leaf senescence, thus improving the photosynthetic capacity (Figure 10A).

MT maintains the RWC of plant leaves during DS [71], thus enhancing stomatal opening and conductivity. This phenomenon contributes to the movement of water and carbon dioxide, thereby improving photosynthesis [72]. In this study, MT maintained a high RWC in cotton leaves during drought-induced senescence (Figure 5A). Moreover, the Gs and Tr of cotton seedlings increased significantly under MT + DS treatment. Previous studies postulate that MT increases stomatal opening by regulating anion channel proteins and dehydrin in the guard cells [73]. MT may thus not regulate stomatal opening to reduce transpiration but may promote root growth to improve the water absorption capacity of the plant (Figure 2E–H). These findings suggest that MT can maintain the RWC and high chlorophyll levels of leaves during drought-induced senescence to preserve the stability and efficiency of the photosynthetic organs.

The response centers of photosynthesis in plants are PSII and PSI. Notably, PSII is extremely sensitive to DS [74,75]. Leaf senescence is often accompanied by decreased PSII activity [76,77]. Studies postulate that leaf senescence can lead to a significant decrease in ΦPSII, Fv/Fm, and ETR and an increase in NPQ [78]. Exogenous MT can alleviate the photoinhibition of PSII and improve the tolerance of plants to abiotic stress [79]. These reports are consistent with the results of this study where there was a significant decrease in ΦPSII, Fv/Fm, and ETR and an increase in NPQ during drought-induced cotton leaf senescence. However, exogenous MT effectively reversed this process and inhibited the decrease of potential NPQ activity during leaf senescence (Figure 4). Fv/Fm reflects the damage degree of photosynthetic equipment during stress [80]. Studies report that exogenous MT effectively increases the Fv/Fm of tomato during leaf senescence [81]. In addition, plants can convert excess light energy into heat energy during the actual production process by inducing NPQ under high light conditions to avoid oxidative damage [82]. However, this photoprotection mechanism can cause photosynthetic loss to some extent [83]. The induction and relaxation of NPQ is the primary mechanism affecting the photosynthetic efficiency of crops [84]. The reduction of the NPQ level by exogenous MT provides an opportunity to further improve the photosynthetic capacity of crops, which helps to delay leaf senescence.

Phenotypically, DS treatment significantly inhibited the growth of cotton seedlings, with the leaves showing symptoms of premature senescence. However, MT + DS treatment alleviated DS-induced growth inhibition. MT promoted the growth of cotton during leaf senescence, embodied by higher aboveground and root morphological indexes (Figure 1 and Figure 2). Similar studies postulate that MT promotes the growth of wheat [85], tomato [86], and tobacco [43] under DS. These findings thus further confirm that MT also phenotypically alleviates drought-induced cotton leaf senescence.

## 4. Materials and Methods

### 4.1. Plant Materials and Growth Conditions

This study used Nongdamian 601, a local commercial cotton (*Gossypium hirsutum* L.) cultivar widely grown in the Yellow River Basin in China. Sterile cotton seeds were germinated in an incubator at 25 °C for 24 h in the dark, sown in plastic pots (11 cm diameter) containing vermiculite, and then maintained in a greenhouse. The growth conditions in the greenhouse were 14/10 h day/night cycle, 28/25 °C (day/night), with a light intensity of 300 μmol m^−2^ s^−1^ during the day, and 45 ± 5% relative humidity. Each pot was irrigated with 100 mL Hoagland nutrient solution (complete nutrient solution) once every two days for one month after the unfolding of the cotyledons.

### 4.2. Experimental Design

The cotton seedlings were treated with MT and drought treatments upon reaching the five-leaf stage: (1) well-watered and 0 μM MT (CK), (2) drought stress and 0 μM MT (DS), (3) drought stress and 100 μM MT (MT + DS). A randomized complete block design was adopted for the experiment, with each treatment replicated thirty times. Seedlings in the CK treatment were irrigated with 100 mL nutrient solution once every two days for 14 days. Seedlings in the DS treatment were irrigated with 100 mL of 15% PEG-6000-containing nutrient solution to induce severe DS in cotton [87]. Seedlings in the MT + DS treatment were sprayed with MT during the dark cycle until the leaves started dripping [88], once every two days for 14 days. This treatment was extremely successful in reducing DS in a prior trial [86,89]. Pot positions were randomly shuffled daily to reduce the positional effects in the artificial climate chamber. The physiological indices of cotton plants were measured from the third leaf (from the top of the plant) at 0, 7, and 14 days after treatment (DAT). In the index measurements, six repeats were employed during non-destructive sampling, while three repeats were employed during destructive sampling.

### 4.3. Plant Growth and Root Morphology

Plant heights were measured using a ruler, while the main stem diameter was determined using Vernier calipers. The green leaf area of the whole plant was calculated using the length and width coefficient method based on ruler measurements [90]. Fresh shoots and roots were weighed using an electric balance, while the dry shoot and root weights were determined following the method of Jiang et al. [62]. The root samples were carefully separated from the growth medium using slow-flowing water. Root length, surface area, diameter, and volume were obtained using the method of Zhu et al. [91].

### 4.4. Evaluation of the Photosynthesis Parameters

The chlorophyll content was measured using SPAD-502 (Konica Minolta, Tokyo, Japan). Leaf photosynthesis parameters, including the net photosynthetic rate (Pn), stomal conductance (Gs), transpiration rate (Tr), and intercellular CO_2_ concentration (Ci), were measured using an LI-6800 photosynthetic instrument (LI-COR, Lincoln, NE, USA). The instrument was set at a light intensity of 300 μmol m^−2^ s^−1^. The light-response curve of photosynthesis was measured at 14 DAT with the light intensity set at 0, 30, 70, 100, 150, 200, 300, 600, 900, 1200, 1500, and 1800 μmol m^−2^ s^−1^. Each treatment was replicated three times and the averages used to fit the light response curve [92]. Chlorophyll fluorescence parameters were measured using a PAM2500 portable chlorophyll fluorometer (Heinz Walz GmbH, Effeltrich, Germany). The parameters included the maximum photochemical efficiency (Fv/Fm) of PSII, the actual photochemical quantum yield of PSII (ΦPSII), non-photochemical quenching (NPQ), and apparent electron transfer efficiency (ETR). There were six repeats in each treatment.

### 4.5. Relative Water Content of the Leaves

The relative water content (RWC) of the leaves was measured following the method of Barrs and Weatherley [93].

### 4.6. The Antioxidant Enzyme Activity and Contents of Reactive Oxygen Species (ROS) and Malondialdehyde (MDA)

The activity of superoxide dismutase (SOD), peroxidase (POD), catalase (CAT), and ascorbate peroxidase (APX), and the contents of hydrogen peroxide (H_2_O_2_), superoxide anions (O_2_^−^), and MDA, were determined following the instructions of the corresponding kits (Nanjing Jiancheng Institute of Biological Engineering, Nanjing, China). 

Cotton leaves were stained with 1 mg/mL 3,3-diaminobenzidine (DAB) and 1 mg/mL nitro-blue tetrazolium (NBT) at 14 DAT to determine the accumulation of H_2_O_2_ and O_2_^−^ in the leaves [94].

### 4.7. Determination of the Hormone Content

Frozen samples (0.2 g) were ground into powder in liquid nitrogen. The contents of auxins (IAA), abscisic acid (ABA), gibberellin acid (GA_3_), and isopentenyladenine (iPA) were subsequently measured using ELISA as described by Duan et al. [95]. The content of endogenous melatonin was determined using the plant melatonin Assay Kit (YJ036336, Shanghai Enzyme-linked Biotechnology Co., Ltd., Shanghai, China).

### 4.8. Transmission Electron Microscope (TEM) Analysis

Cotton leaves were sampled at 14 DAT and were subsequently sliced and prepared as described by Jiang et al. [62]. The leaf slices were subsequently examined using a transmission electron microscope (FEI Tecnai Spirit 120kv, FEI Company, Hillsborough, OR, USA).

### 4.9. qRT-PCR

Total plant RNA was isolated using the FastPure Plant Total RNA isolation Kit (Vazyme, Nanjing, China), followed by cDNA synthesis using the HiScript^®^ Ⅱ Q RT SuperMix for qPCR Kit (Vazyme, Nanjing, China). Primers were designed using the Primer5.0 software (Primer Premier, Oakville, ON, Canada), and their specificity was detected using the Primer-BLAST tool on NCBI. The primers were synthesized by Sangon Biotech Co., Ltd. (Shanghai, China) and their sequences are outlined in Table 2. The gene expression level per treatment was the average of three biological replicates.

### 4.10. Statistical Analysis

IBM SPSS Statistics 26.0 software (IBM Corp, Armonk, NY, USA) was used for one-way analysis of variance (ANOVA) among the treatment groups. Figures were drawn using GraphPad Prism 8.0 (Graph Pad Software, San Diego, CA, USA) and Origin Pro 2022b (OriginLab, Northampton, MA, USA) based on the statistical analysis results.

## 5. Conclusions

This study revealed that spraying MT can increase the content of endogenous melatonin, coordinate the up-regulation of IAA, GA_3_, and iPA, inhibit ABA synthesis and chlorophyll degradation, and inhibit the expression of senescence genes, thus delaying leaf senescence (Figure 11). MT also enhanced the antioxidant enzyme activity, decreased the membrane lipid peroxidation level and excessive accumulation of ROS, and reduced H_2_O_2_ and O_2_^−^ by 33.44% and 37.68%, respectively, thus alleviating the inhibition of DS-induced leaf senescence on the growth of cotton seedlings. Chloroplast ultrastructure analysis further demonstrated that MT protected the integrity of the chloroplast lamellar structure during drought-induced leaf senescence. The reduction of oxidative damage in cotton leaves can be attributed to the maintenance of higher chlorophyll content and RWC and increased photosynthetic capacity. Moreover, exogenous MT delayed leaf senescence by inhibiting the expression of senescence genes, such as *GhWRKY27/71* and *GhNAC12*. The morphological, physiological, biochemical, and chloroplast ultrastructure analysis findings of this study collectively highlight that spraying MT can potentially delay drought-induced cotton leaf senescence.

## Figures and Tables

**Figure 1 ijms-24-07269-f001:**
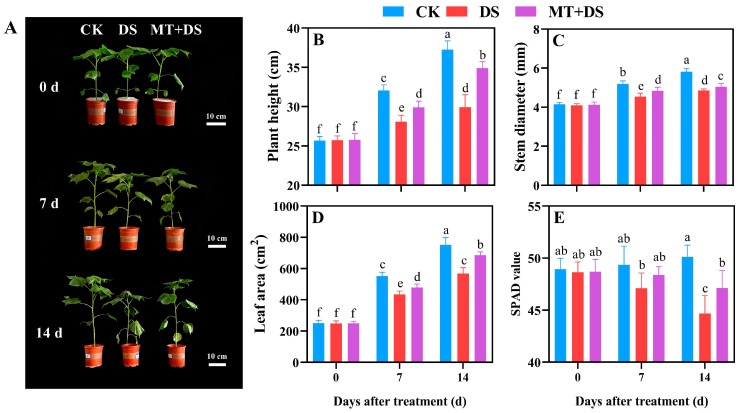
Photos and bar graphs showing the effects of melatonin on the phenotype (**A**), plant height (**B**), stem diameter (**C**), leaf area (**D**), and SPAD value (**E**) of cotton seedlings during drought-induced leaf senescence. Results are presented as means ± standard error, n = 6; significant differences are denoted by lowercase letters (*p* < 0.05), according to Duncan’s new multiple range test. CK denotes well-watered + 0 μM MT treatment; DS denotes drought stress + 0 μM MT treatment; MT + DS denotes drought stress + 100 μM MT treatment.

**Figure 2 ijms-24-07269-f002:**
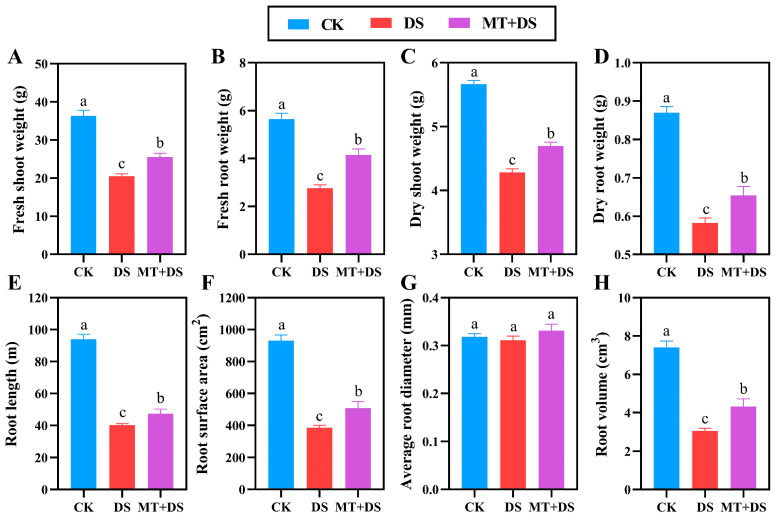
Bar graphs showing the effects of melatonin on the fresh shoot weight (**A**), fresh root weight (**B**), dry shoot weight (**C**), dry root weight (**D**), root length (**E**), root surface area (**F**), average root diameter (**G**), and root volume (**H**) of cotton seedlings during drought-induced leaf senescence. Results are presented as means ± standard error, n = 3; significant differences are denoted by lowercase letters (*p* < 0.05), according to Duncan’s new multiple range test. CK denotes well-watered + 0 μM MT treatment; DS denotes drought stress + 0 μM MT treatment; MT + DS denotes drought stress + 100 μM MT treatment.

**Figure 3 ijms-24-07269-f003:**
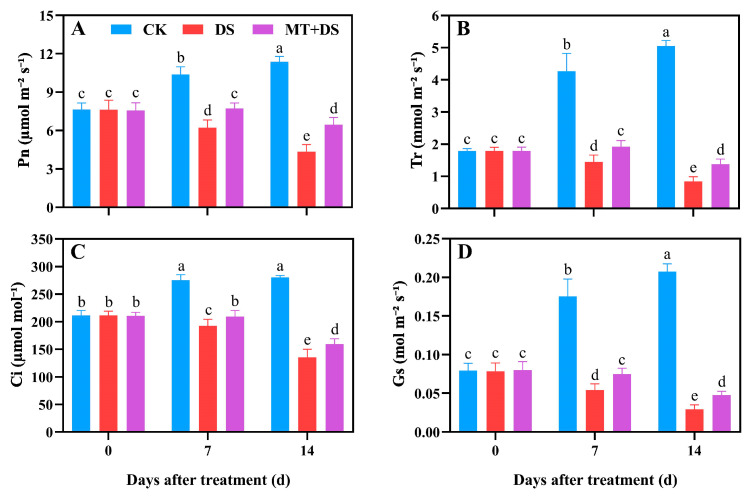
Bar graphs showing the effects of melatonin on the net photosynthetic rate (Pn; **A**), transpiration rate (Tr; **B**), intercellular CO_2_ concentration (Ci; **C**), and stomatal conductance (Gs; **D**) of cotton seedlings during drought-induced senescence. Results are presented as means ± standard error, n = 6; significant differences are denoted by lowercase letters (*p* < 0.05), according to Duncan’s new multiple range test. CK denotes well-watered + 0 μM MT treatment; DS denotes drought stress + 0 μM MT treatment; MT + DS denotes drought stress + 100 μM MT treatment.

**Figure 4 ijms-24-07269-f004:**
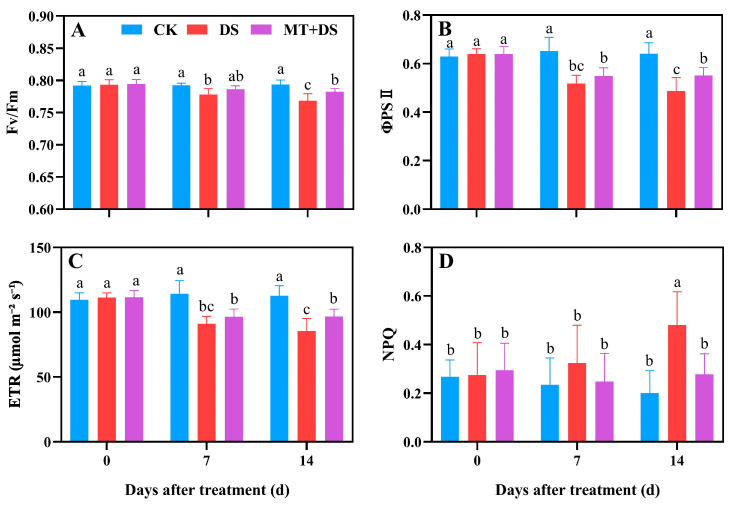
Bar graphs showing the effects of melatonin on the maximum photochemical efficiency (Fv/Fm; **A**), PSII actual photochemical quantum yield (ΦPSII; **B**), electron transfer efficiency (ETR; **C**), and nonphotochemical quenching (NPQ; **D**) of cotton seedlings during drought-induced senescence. Results are presented as means ± standard error, n = 6; significant differences are denoted by lowercase letters (*p* < 0.05), according to Duncan’s new multiple range test. CK denotes well-watered + 0 μM MT treatment; DS denotes drought stress + 0 μM MT treatment; MT + DS denotes drought stress + 100 μM MT treatment.

**Figure 5 ijms-24-07269-f005:**
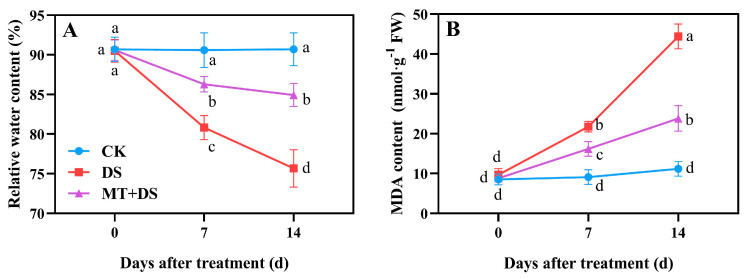
Line graphs showing the effects of melatonin on the relative water content (RWC; **A**) and malonaldehyde (MDA; **B**) contents of cotton leaves during drought-induced senescence. Results are presented as means ± standard error, n = 3; significant differences are denoted by lowercase letters (*p* < 0.05), according to Duncan’s new multiple range test. CK denotes well-watered + 0 μM MT treatment; DS denotes drought stress + 0 μM MT treatment; MT + DS denotes drought stress + 100 μM MT treatment.

**Figure 6 ijms-24-07269-f006:**
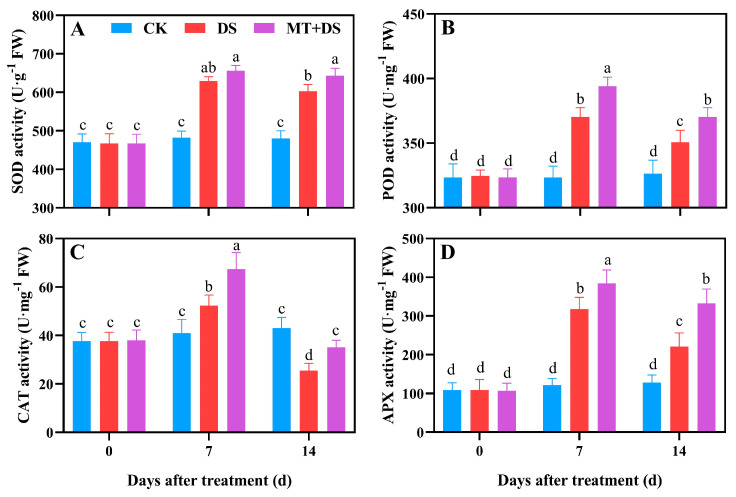
Bar graphs showing the effects of melatonin on the superoxide dismutase (SOD; **A**), peroxidase (POD; **B**), catalase (CAT; **C**), and ascorbate peroxidase (APX; **D**) activity of cotton seedlings during drought-induced senescence. Results are presented as means ± standard error, n = 6; significant differences are denoted by lowercase letters (*p* < 0.05), according to Duncan’s new multiple range test. CK denotes well-watered + 0 μM MT treatment; DS denotes drought stress + 0 μM MT treatment; MT + DS denotes drought stress + 100 μM MT treatment.

**Figure 7 ijms-24-07269-f007:**
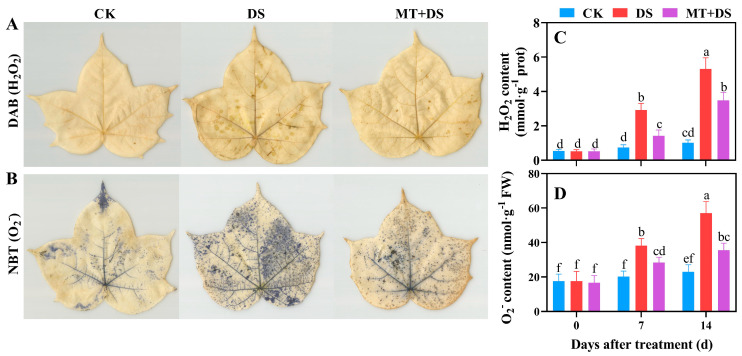
Leaf photos and bar graphs showing the effect of melatonin on the contents of ROS in the leaves of cotton seedlings during drought-induced senescence. (**A**) Micrograph showing DAB staining of hydrogen peroxide (H_2_O_2_) in cotton plants; (**B**) micrograph showing NBT staining of superoxide anion (O_2_^−^) in cotton plants; (**C**) H_2_O_2_ contents; (**D**) O_2_^−^ contents. Results are presented as means ± standard error, n = 3; significant differences are denoted by lowercase letters (*p* < 0.05), according to Duncan’s new multiple range test. CK denotes well-watered + 0 μM MT treatment; DS denotes drought stress + 0 μM MT treatment; MT + DS denotes drought stress + 100 μM MT treatment.

**Figure 8 ijms-24-07269-f008:**
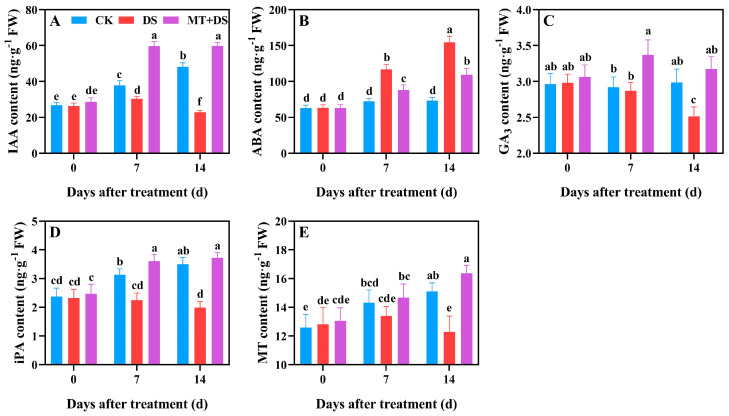
Bar graphs showing the effects of melatonin on the auxin (IAA; **A**), abscisic acid (ABA; **B**), gibberellin acid (GA_3_; **C**), isopentenyladenine (iPA; **D**), and endogenous melatonin (MT; **E**) contents of cotton seedlings during drought-induced senescence. Results are presented as means ± standard error, n = 3; significant differences are denoted by lowercase letters (*p* < 0.05), according to Duncan’s new multiple range test. CK denotes well-watered + 0 μM MT treatment; DS denotes drought stress + 0 μM MT treatment; MT + DS denotes drought stress + 100 μM MT treatment.

**Figure 9 ijms-24-07269-f009:**
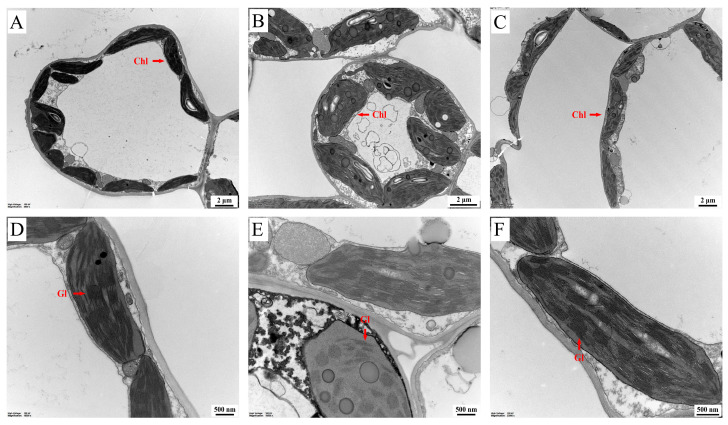
Transmission electron micrographs (TEM) of mesophyll cells of the cotton leaf. (**A**–**C**) structure of mesophyll cells of plants under CK, DS, and MT + DS treatments, respectively; (**D**–**F**) the relatively high magnified view of chloroplasts of plants under CK, DS, and MT + DS treatments, respectively. CK denotes well-watered + 0 μM MT treatment; DS denotes drought stress + 0 μM MT treatment; MT + DS denotes drought stress + 100 μM MT treatment. *Chl*; chloroplast; *Gl*: Grana lamellae.

**Figure 10 ijms-24-07269-f010:**
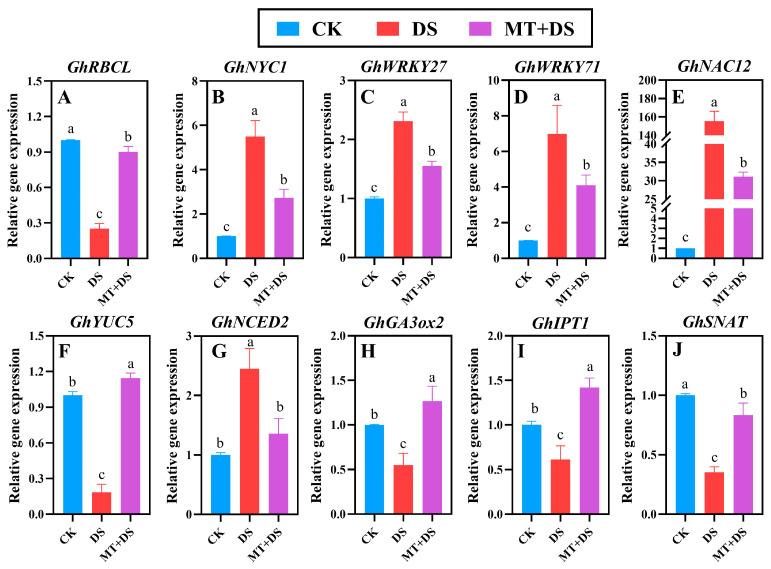
Bar graphs showing the effects of melatonin on the transcript levels of chlorophyll degradation genes, senescence-related genes, and hormone synthesis genes during leaf senescence. (**A**,**B**) Transcript levels of chlorophyll degrading genes *large subunit of Rubisco, ribulose-1, 5-bisphosphate carboxylase/oxygenase* (*GhRBCL*) and *non-yellow coloring 1* (*GhNYC1*); (**C**–**E**) transcript levels of leaf senescence-related genes *GhWRKY27/71* and *GhNAC12*; (**F**–**J**) transcript levels of hormone synthesis genes *flavin-binding monooxygenase family protein* (*GhYUC5*), *nine-cis-epoxycarotenoid dioxygenase 2* (*GhNCED2*), *gibberellin 3-oxidase 2* (*GhGA3ox2*), *tRNAisopentenyltransferase 2* (*GhIPT2*), and *serotonin N-acetyltransferase* (*GhSNAT*). Results are presented as means ± standard error, n = 3; significant differences are denoted by lowercase letters (*p* < 0.05), according to Duncan’s new multiple range test. CK denotes well-watered + 0 μM MT treatment; DS denotes drought stress + 0 μM MT treatment; MT + DS denotes drought stress + 100 μM MT treatment.

**Figure 11 ijms-24-07269-f011:**
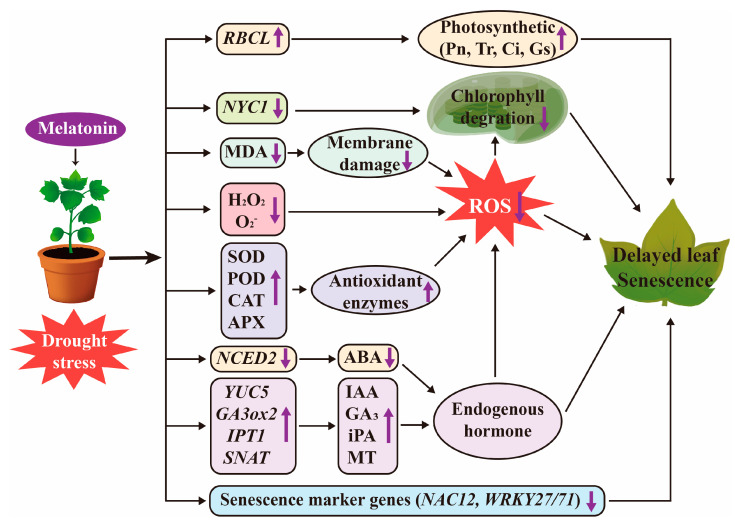
Mechanistic model of melatonin alleviating cotton leaf senescence under drought stress.

**Table 1 ijms-24-07269-t001:** Effect of melatonin on the fitting parameters of the light response curve of cotton seedlings during drought-induced senescence.

Treatment	P_max_ (μmol/m^2^ s^−1^)	LSP (μmol/m^2^ s^−1^)	LCP (μmol/m^2^ s^−1^)	R_d_ (μmol/m^2^ s^−1^)	AQE (μmol/mol)	*R*^2^ of Model Fitting
CK	20.68 ± 3.94 a	1761.44 ± 227.26 a	50.64 ± 7.6 a	3.18 ± 0.5 a	0.0688 ± 0.0015 a	0.9995
DS	12.84 ± 1.85 b	1447.5 ± 179.85 a	16.63 ± 3.06 c	0.82 ± 0.05 c	0.0526 ± 0.0123 a	0.9984
MT + DS	17.43 ± 0.27 ab	1637.41 ± 68.71 a	33.21 ± 4.39 b	2.08 ± 0.24 b	0.0675 ± 0.0013 a	0.9995

Data are presented as means ± standard error (n = 3); significant differences are denoted by lowercase letters. Abbreviations and symbols: P_max_: the maximum net photosynthetic rate; LSP: light saturation point; LCP: light compensation point; R_d_: respiratory rate; AQE: the apparent quantum efficiency of net carbon assimilation. CK denotes well-watered + 0 μM MT treatment; DS denotes drought stress + 0 μM MT treatment; MT + DS denotes drought stress + 100 μM MT treatment.

**Table 2 ijms-24-07269-t002:** The primer pairs used in this experiment.

Gene Name	Forward Primer (5′ to 3′)	Reverse Primer (5′ to 3′)
*GhRBCL*	TTCCGAGTAACTCCTCAACCC	CTTACAGATGCACCGCCCG
*GhNYC1*	TGCTTGTGGTGGGCTGCTA	GTCAGCGAAAGTACGCACAA
*GhWRKY27*	CCTACGGAACAGCCACAACAA	CTTCCAGCAGGTTTTCAGAGC
*GhWRKY71*	ACTTGGGTGGATACGACTCTC	TCCTGCTCACCAATTCCATGT
*GhNAC12*	CAAAAGAGTCCCAATGGCAA	CTGCTTTACGGTCTTCTATCGG
*GhYUC5*	ACTCCTGCTTTTGTTCTTCTT	CTTGTGAAACCGTCTCGTT
*GhNCED2*	TACGATGTTATCCAAAAGCC	GAAAATAGAGTCGGGAGGTG
*GhGA3ox2*	AACTTAAAGCTGCAAGGTCTC	ACTCCTAATCTACTTCGTGGG
*GhIPT1*	CAATCGCACGCCTAATCCCT	GCATTGGTGAATGATGACCCT
*GhSNAT*	CACAAATCGTTGAACCACC	AATTAGCCTCTTTTGCTCATT
*ACTIN14*	ATCCTCCGTCTTGACCTTG	TGTCCGTCAGGCAACTCAT

## Data Availability

Not applicable.

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
