# Peer review of "Morphological and Physiological Mechanisms of Melatonin on Delaying Drought-Induced Leaf Senescence in Cotton"

_ijms, 2023, doi:10.3390/ijms24087269_

Round 1

Reviewer 1 Report

The manuscript entitled "Morphological and Physiological Mechanisms of Melatonin on Delaying Drought-Induced Leaf Senescence in Cotton" aimed at aimed to explore the morphological and physiological mechanisms of MT on delaying drought-induced leaf senescence in cotton seedlings. They claimed that MT can effectively activate the antioxidant enzyme system, improve photosynthetic efficiency, reduce chlorophyll degradation and ROS accumulation, and inhibit ABA synthesis, thereby delaying drought-induced leaf senescence in cotton. The subject is interesting, and was written well. Following are the several minor points needing to be modified or solved carefully:

1. In Abstract, line 19, change “This study aimed to explore the effect and the morphophysiological and physiological mechanisms of MT on delaying drought-induced leaf senescence in cotton seedlings” to “This study aimed to explore the effect of MT on delaying drought-induced leaf senescence in cotton seedlings, and to clarify its morphological and physiological mechanism”.

2. Why is there no mention of morphological results in the abstract, but your title is “Morphological and Physiological Mechanisms of Melatonin on Delaying Drought-Induced Leaf Senescence in Cotton”?

3. Line 25, delete “upregulated expression of auxins (IAA), gibberellin acid (GA3), and isopentenyladenine (iPA) synthesis genes, significant down-regulation of chlorophyll degradation-related genes (GhNYC1) and senescence marker genes (GhNAC12 and GhWRKY27/71)”, or adjust the position of this sentence.

4. Line 31, change “MT can effectively activate the antioxidant enzyme system” to “MT can effectively enhance the antioxidant enzyme system”. Rewrite this sentence, “The findings of this study collectively suggest that MT can effectively activate the antioxidant enzyme system, improve photosynthetic efficiency, reduce chlorophyll degradation and ROS accumulation, and inhibit ABA synthesis, thereby delaying drought-induced leaf senescence in cotton”. This is more like a repetition of the above results.

5. In Introduction, line 74, change “Exploring methods to delay leaf senescence are thus prerequisites for maintaining and enhancing agricultural production” to “Exploring methods to delay leaf senescence are thus prerequisites for maintaining and enhancing agricultural production under abiotic stress conditions”.

6. Line 79, add a space after “root development”. Check the spaces in the article

7. Line 96, change “anatomical structure” to “ultra-microstructure”.

8. In Results, line 106, add “(DS)” after “drought stress”.

9. Line 114, in Figure 1, delete “d” after the numbers of abscissa. The same below.

10. Line 170, change “non-photochemical quenching” to “non-photochemical quenching”. Check the font format in the article.

11. Line 221-222, please list the full names of DAB and NBT that appear for the first time in the article.

12. In Discussion, line 383, why put the relevant content of expression of senescence marker genes here?

13. Line 409, rewrite this sentence, “However, MT+DS treatment also reduced the damage to the chloroplast structure”.

14. Line 417, “Moreover, there was a significant increase in MDA, H2O2, and O2- under MT+DS treatment compared to the control.” I don't think it's appropriate to put such a sentence here.

15. Line 433, I think it is more appropriate to discuss the 3.2 and 3.4 together.

16. In Materials and Methods, line 489, delete “for the various assays”.

17. Line 509, delete “At least”.

18. Line 525, add a space between reference numbers. Please check the entire manuscript.

19. Line 528, further clarify the number of repetitions of the measurement indicators, or add to the statistical analysis section

20. Please check all the reference formats.

21. English language needs further polishing.

Author Response

Dear Reviewer

On behalf of my co-authors, we thank you very much for giving us an opportunity to revise our manuscript again, we appreciate reviewers very much for their positive and constructive comments and suggestions on our manuscript entitled “Morphological and Physiological Mechanisms of Melatonin on Delaying Drought-Induced Leaf Senescence in Cotton”. Those comments are all valuable and very helpful for revising and improving our paper, as well as the important guiding significance to us researches. We have studied comments carefully and have made correction which we hope meet with approval. Revised portion are marked in red in the paper.

 The main corrections in the paper and the responds to your comments are as flowing:

  1. In Abstract, line 19, change “This study aimed to explore the effect and the morphophysiological and physiological mechanisms of MT on delaying drought-induced leaf senescence in cotton seedlings” to “This study aimed to explore the effect of MT on delaying drought-induced leaf senescence in cotton seedlings, and to clarify its morphological and physiological mechanism”.

Response: Thanks for your suggestion, we have changed “This study aimed to explore the effect and the morphophysiological and physiological mechanisms of MT on delaying drought-induced leaf senescence in cotton seedlings” to “This study aimed to explore the effect of MT on delaying drought-induced leaf senescence in cotton seedlings, and to clarify its morphological and physiological mechanism”.

  1. Why is there no mention of morphological results in the abstract, but your title is “Morphological and Physiological Mechanisms of Melatonin on Delaying Drought-Induced Leaf Senescence in Cotton”?

Response: Thank you for your suggestion. In the abstract, we focus on systematically explaining how melatonin alleviates leaf senescence physiologically, but we don’t mention too much about the morphological results because of the lexical limitations of the abstract. The results and analysis show the morphological results in a very comprehensive way.

  1. Line 25, delete “upregulated expression of auxins (IAA), gibberellin acid (GA3), and isopentenyladenine (iPA) synthesis genes, significant down-regulation of chlorophyll degradation-related genes (GhNYC1) and senescence marker genes (GhNAC12and GhWRKY27/71)”, or adjust the position of this sentence.

Response: Thank you for your suggestion, we have changed the position of this sentence and simplified it.

  1. Line 31, change “MT can effectively activate the antioxidant enzyme system” to “MT can effectively enhance the antioxidant enzyme system”. Rewrite this sentence, “The findings of this study collectively suggest that MT can effectively activate the antioxidant enzyme system, improve photosynthetic efficiency, reduce chlorophyll degradation and ROS accumulation, and inhibit ABA synthesis, thereby delaying drought-induced leaf senescence in cotton”. This is more like a repetition of the above results.

Response: Thank you for your suggestion, we have rewritten this sentence according to your suggestion: “The findings of this study collectively suggest that MT can effectively activate the antioxidant enzyme system, improve photosynthetic efficiency, reduce chlorophyll degradation and ROS accumulation, and inhibit ABA synthesis, thereby delaying drought-induced leaf senescence in cotton”.

  1. In Introduction, line 74, change “Exploring methods to delay leaf senescence are thus prerequisites for maintaining and enhancing agricultural production” to “Exploring methods to delay leaf senescence are thus prerequisites for maintaining and enhancing agricultural production under abiotic stress conditions”.

Response: Thanks for your suggestion, we have changed “Exploring methods to delay leaf senescence are thus prerequisites for maintaining and enhancing agricultural production” to “Exploring methods to delay leaf senescence are thus prerequisites for maintaining and enhancing agricultural production under abiotic stress conditions”.

  1. Line 79, add a space after “root development”.Check the spaces in the article.

Response: Thanks for your suggestion, we have added spaces after root as you suggested and checked the full text.

  1. Line 96, change “anatomical structure” to “ultra-microstructure”.

Response: Thanks for your suggestion, we have changed “anatomical structure” to “ultra-microstructure”.

  1. In Results, line 106,add “(DS)” after “drought stress”.

Response: Thanks for your suggestion, we have added “(DS)” after “drought stress”.

  1. Line 114, in Figure 1, delete “d” after the numbers of abscissa. The same below.

Response: Thanks for your suggestion, we have deleted “d” after the numbers of abscissa in Figures1, 3, 4, 5, 6, 7, 8.

  1. Line 170, change “non-photochemical quenching” to “non-photochemical quenching”.Check the font format in the article.

Response: Thanks for your suggestion, we have changed “non-photochemical quenching” to “non-photochemical quenching”. Check the font format in the article.

  1. Line 221-222, please list the full names of DAB and NBT that appear for the first time in the article.

Response: Thanks for your suggestion, we have changed “DAB and NBT” to “3,3-diaminobenzidine (DAB) and nitro-blue tetrazolium (NBT)”.

  1. In Discussion,line 383, why put the relevant content of expression of senescence marker genes here?

Response: Thank you for your suggestion. We have revised the title according to the content contained in 3.1. Modify the title of 3.1 to: “MT delays leaf senescence by regulating phytohormone biosynthesis and senescence marker genes expression.”

  1. Line 409, rewrite this sentence, “However, MT+DS treatment also reduced the damage to the chloroplast structure”.

Response: Thank you for your suggestion, we have changed “However, MT+DS treatment also reduced the damage to the chloroplast structure” to “However, under MT+DS treatment, the damage of chloroplast caused by drought stress was significantly reduced.”

  1. Line 417, “Moreover, there was a significant increase in MDA, H2O2, and O2- under MT+DS treatment compared to the control.” I don't think it's appropriate to put such a sentence here.

Response: Thank you for your suggestion, we have changed “Moreover, there was a significant increase in MDA, H2O2, and O2- under MT+DS treatment compared to the control.” I don't think it's appropriate to put such a sentence here.” to “Moreover, there was a significant increase in MDA, H2O2, and O2- under MT+DS treatment compared to DS treatment.”

  1. Line 433, I think it is more appropriate to discuss the 3.2 and 3.4 together.

Response: Thank you very much for your suggestion. We also think that the contents of 3.2 and 3.4 are relatively close, but the author would like to explore the mechanism of melatonin in depth from the two aspects of chloroplast degradation and photosynthetic efficiency improvement.

  1. In Materials and Methods, line 489,delete “for the various assays”.

Response: Thanks for your suggestion, we have deleted “for the various assays”.

  1. Line 509, delete “At least”.

Response: Thanks for your suggestion, we have deleted “At least”.

  1. Line 525, add a space between reference numbers. Please check the entire manuscript.

Response: Thanks for your suggestion, we have added spaces and conducted troubleshooting throughout the entire article.

  1. Line 528, further clarify the number of repetitions of the measurement indicators, or add to the statistical analysis section.

Response: Thanks for your suggestion, we have further clarified the number of repetitions of the measurement indicators.

  1. Please check all the reference formats.

Response: Thanks for your suggestion, we have checked the format of the reference again.

  1. English language needs further polishing.

Response: Thanks for your suggestions, we have made further polish to the English language.

We tried our best to improve the manuscript and made some changes in the manuscript. These changes will not influence the content and framework of the paper. We did not list the changes but marked them in red, including the language modifications, in the revised paper.

We appreciate the wonderful work, and we hope that the corrections will meet with approval.

Once again, we thank you very much for the comments and suggestions.

Sincerely Yours,

Professor Cundong Li and Liantao Liu

College of Agronomy

Hebei Agricultural University,

No. 2596 South Lucky Road, Baoding

Hebei P.R. China.

Tel: +86 (0)312 7520660.

E-mail: nxylcd@hebau.edu.cn and liult@hebau.edu.cn

Reviewer 2 Report

The manuscript presented by Yang and colleagues is an interesting study on the effect of melatonin on protecting cotton plants to the drought stress and delaying leaf senescence. Authors were able to describe the melatonin putative mechanism of  action by physiological and phenotypic analysis, hormones quantification and gene expression analysis of key marker genes.

In my opinion, there are just few things that can be improved:

Fig 1A: add a scale bar.

Page 13:  please, indicate full gene name. Eg. non-yellow coloring 1 (NYC1). Extend to all abbreviations when are used for first time are used.

Line 502: 100 ml…

Author Response

Dear Reviewer

On behalf of my co-authors, we thank you very much for giving us an opportunity to revise our manuscript again, we appreciate reviewers very much for their positive and constructive comments and suggestions on our manuscript entitled “Morphological and Physiological Mechanisms of Melatonin on Delaying Drought-Induced Leaf Senescence in Cotton”. Those comments are all valuable and very helpful for revising and improving our paper, as well as the important guiding significance to us researches. We have studied comments carefully and have made correction which we hope meet with approval. Revised portion are marked in red in the paper.

 The main corrections in the paper and the responds to your comments are as flowing:

  1. Fig 1A: add a scale bar.

Response: Thank you for your suggestion. We have added a scale to figure 1A.

  1. Page 13:  please, indicate full gene name. Eg. non-yellow coloring 1 (NYC1). Extend to all abbreviations when are used for first time are used.

Response: Thanks for your suggestion, we have made famous the full name of the gene in the article and extended to all abbreviations when are used for first time are used.

  1. Line 502: 100 ml

Response: Thanks for your suggestion, we added a space after 100.

We tried our best to improve the manuscript and made some changes in the manuscript. These changes will not influence the content and framework of the paper. We did not list the changes but marked them in red, including the language modifications, in the revised paper.

We appreciate the wonderful work, and we hope that the corrections will meet with approval.

Once again, we thank you very much for the comments and suggestions.

Sincerely Yours,

Professor Cundong Li and Liantao Liu

College of Agronomy

Hebei Agricultural University,

No. 2596 South Lucky Road, Baoding

Hebei P.R. China.

Tel: +86 (0)312 7520660.

E-mail: nxylcd@hebau.edu.cn and liult@hebau.edu.cn

Reviewer 3 Report

Paper subsections must be properly placed with respect to current journal standards

findings can be presented in more elloborate way

Author Response

Dear Reviewer

On behalf of my co-authors, we thank you very much for giving us an opportunity to revise our manuscript again, we appreciate reviewers very much for their positive and constructive comments and suggestions on our manuscript entitled “Morphological and Physiological Mechanisms of Melatonin on Delaying Drought-Induced Leaf Senescence in Cotton”. Those comments are all valuable and very helpful for revising and improving our paper, as well as the important guiding significance to us researches. We have studied comments carefully and have made correction which we hope meet with approval. Revised portion are marked in red in the paper.

 The main corrections in the paper and the responds to your comments are as flowing:

  1. Line28, Commented [A1]: Quantify the results.

Response: Thanks for your suggestion, we have quantified the result and finally modified it to: " The delay was embodied by the increased chlorophyll content, photosynthetic capacity, and antioxidant enzyme activities, and decreased the contents of H2O2, O2, and abscisic acid (ABA) by 34.44%, 37.68%, and 29.32%, respectively.”

  1. Line 95, Commented [A2]: Support with refrence

Response: Thanks for your suggestion, we have cited the following references.

Zhao, Y.-Q.; Zhang, Z.-W.; Chen, Y.-E.; Ding, C.-B.; Yuan, S.; Reiter, R.J.; Yuan, M. Melatonin: A Potential Agent in Delaying Leaf Senescence. Crit. Rev. Plant Sci. 2021, 40, 1–22, doi:10.1080/07352689.2020.1865637.

  1. Line 104, Commented [A3]: Mention the work protocol and methodology and base materialused

Response: Thanks for your suggestion, we introduced the work protocol and methodology and base materialused: “Cotton seedlings were irrigated with PEG-6000 for 14 days to simulate drought stress to induce leaf senescence.”

  1. Line 128, Commented [A4]: Avoid presumption support with valid refrences

Response: Thanks for the suggestion, we have replaced "could" with "can".

  1. Line 142, Commented [A5]: rewrite

Response: Thanks for your suggestion, we have changed “Exploring methods to delay leaf senescence are thus prerequisites for maintaining and enhancing agricultural production” to “Exploring methods to delay leaf senescence are thus prerequisites for maintaining and enhancing agricultural production under abiotic stress conditions”.

  1. Line 530, Commented [A6]: role of Fv/Fm ratio elloborate

Response: Thanks for your suggestion. Fv/Fm is the maximum quantum yield of PSⅡ under dark adaptation, which reflects the maximum light conversion efficiency of the PSⅡ reaction center and can effectively represent the stress degree path of leaves.

  1. Line 578, Commented [A7]: quanitify the ROS accumulation in refrence to drought effects

Response: Thanks for your suggestion. We quantify the ROS accumulation in refrence to drought effects and rewrite the original sentence as follows: "MT also enhanced the antioxidant enzymes’ activity and decreased membrane lipid peroxidation level and excessive accumulation of ROS, and reduce H2O2 and O2 by 33.44% and 37.68%, respectively, thus alleviating the inhibition of drought stress-induced leaf senescence on the growth of cotton seedlings.”

We tried our best to improve the manuscript and made some changes in the manuscript. These changes will not influence the content and framework of the paper. We did not list the changes but marked them in red, including the language modifications, in the revised paper.

We appreciate the wonderful work, and we hope that the corrections will meet with approval.

Once again, we thank you very much for the comments and suggestions.

Sincerely Yours,

Professor Cundong Li and Liantao Liu

College of Agronomy

Hebei Agricultural University,

No. 2596 South Lucky Road, Baoding

Hebei P.R. China.

Tel: +86 (0)312 7520660.

E-mail: nxylcd@hebau.edu.cn and liult@hebau.edu.cn

Round 2

Reviewer 3 Report

The authors have submitted the revised version and manuscript is fit for publication